# Endometriosis and Isthmocele: Common or Rare?

**DOI:** 10.3390/jcm11051158

**Published:** 2022-02-22

**Authors:** Marietta Gulz, Sara Imboden, Konstantinos Nirgianakis, Franziska Siegenthaler, Tilman T. Rau, Michael D. Mueller

**Affiliations:** 1Department of Gynecology and Obstetrics, Bern University Hospital, University of Bern, 3010 Bern, Switzerland; sara.imboden@insel.ch (S.I.); konstantinos.nirgianakis@insel.ch (K.N.); franziska.siegenthaler@insel.ch (F.S.); michael.mueller@insel.ch (M.D.M.); 2Institute of Pathology, Bern University Hospital, University of Bern, 3010 Bern, Switzerland; tilman.rau@pathology.unibe.ch

**Keywords:** isthmocele, uterine scar defect, laparoscopic isthmocele repair, rendezvous technique, endometriosis, infertility

## Abstract

Higher cesarean section rates and better ultrasound diagnostics have led to a more frequent diagnosis of isthmocele, a cesarean scar defect. Sometimes, endometriosis is found in the isthmocele, but simultaneous extrauterine endometriosis and endometriosis in the isthmocele have not yet been reported. Additionally, the surgical technique to repair the isthmocele is the subject of ongoing controversy. The aim of this study is to analyze a possible correlation between uterine scar (isthmocele) endometriosis and extrauterine endometriosis and to investigate the outcome of laparoscopic isthmocele resection in the rendezvous technique. In this single-center retrospective study, we included 83 women of reproductive age with symptomatic isthmocele undergoing laparoscopic isthmocele repair in rendezvous technique from 2004 to 2020 at the University of Bern. We collected data on patient and surgical characteristics as well as on postoperative outcomes (symptoms, further pregnancy, and pregnancy outcomes) retrospectively. We analyzed and compared these data for patients with and without endometriosis. Endometriosis was diagnosed during surgery in 22 out of 83 operated patients (26.5%). Diagnosis of isthmocele endometriosis (*n* = 9, 11%) was significantly higher in patients with extrauterine endometriosis (*n* = 6, *p* = 0.004). While the duration of surgery was significantly longer for patients with endometriosis (*p* = 0.006), the groups did not differ with regard to blood loss or complications. In addition, both groups showed similar indications for isthmocele repair (infertility, abnormal uterine bleeding, or dysmenorrhea). Surgery significantly improved abnormal uterine bleeding (χ^2^ *p* < 0.001), dysmenorrhea (χ^2^, *p* = 0.03), and infertility (χ^2^, *p* < 0.001). Regardless of the presence of endometriosis, 25 of 40 (63%) infertile patients became pregnant after surgery. In one out of eight pregnancies, however, we observed scar complications during pregnancy such as uterine scar pregnancy (*n* = 3), uterine scar dehiscence (*n* = 3), and placenta previa (*n* = 1). Endometriosis is a non-negligible intraoperative finding in patients with symptomatic isthmocele. The laparoscopic approach in the rendezvous technique is safe and effective. Therefore, this method should be recommended, especially in women with secondary infertility, and preoperatively simultaneous endometriosis resection should be discussed with the patient. In follow-up, postoperative pregnancies have to be monitored with care.

## 1. Introduction

Isthmocele, also called “uterine scar defect”, “pouch”, or “niche”, is a common long-term consequence of cesarean section, which can severely impact the quality of life. Higher rates of cesarean section—one of the most frequent operations worldwide—and better ultrasound diagnostics have led to more frequent diagnoses of isthmocele. The prevalence of isthmocele ranges from 24% to 88% [1,2].

The pathogenesis and risk factors are not yet fully understood. Repeated cesarean section and several preoperative, intraoperative, and individual factors play a role [1,3,4,5,6]. Another potential risk factor is endometriosis. Isthmocele and endometriosis show similar symptoms: infertility, bleeding disorders, and pain. It is known that isthmocele is a consequence of cesarean section. Endometriosis could also be caused by this procedure, as endometrial tissue spreads into the abdomen during surgery [7]. Several studies report the existence of endometriosis in the isthmocele (also called uterine scar endometriosis), but the presence of coexisting extrauterine endometriosis has not yet been reported [8,9,10,11]. Just one study presents findings concerning extrauterine endometriosis in patients with symptomatic isthmocele [12]. The present study sheds light on the relationship between isthmocele and endometriosis by investigating two questions: (a) whether endometriosis, both extrauterine endometriosis and uterine scar endometriosis, is a frequent finding in patients with isthmocele and (b) whether both types of endometriosis are correlated.

Moreover, we focus on the surgical treatment of isthmocele, investigating the outcome of laparoscopic isthmocele resection in the rendezvous technique. The rendezvous technique is the simultaneous application of laparoscopy and hysteroscopy to detect and treat isthmocele [13].

In general, laparoscopic isthmocele repair is considered an effective method for achieving pregnancy in secondary infertile women and for improving bleeding disorders and pain [8,14]. However, the literature does not provide sufficient evidence on pregnancy outcomes and complications after laparoscopic isthmocele resection in the rendezvous technique. Here, we evaluate the outcomes of this approach.

## 2. Materials and Methods

In this retrospective study, we investigate the data of patients with a symptomatic isthmocele who were treated at the SEF (Stiftung Endometriose Forschung) certified endometriosis center of Bern University Hospital between April 2004 and March 2020. A total of 111 patients were identified. Of these patients, 83 underwent laparoscopic isthmocele resection in the rendezvous technique at the Department of Gynecology and Obstetrics, Bern University Hospital. All the patients were counseled on the excision of coexisting findings such as cysts, endometriosis, and adhesions.

### 2.1. Performed Surgery

Laparoscopic isthmocele repair in the rendezvous technique is a minimally invasive method combining laparoscopy and hysteroscopy. This procedure was performed as described in our previous study [13]: after dissection of the bladder from the cervix and uterus, the isthmocele was identified by hysteroscopic transillumination. The defect was excised using an electrosurgical hook. Then, hysterotomy closure was performed applying the single-layer technique using single-knot sutures. The knots were tied extracorporeally. The suture material was nonabsorbable and made of polyester fibers (PremiCron^®^ (B.Braun, Melsungen, Germany) or Ethibond^®^ (Ethicon, Bridgewater, NJ, USA)). Optionally, the bladder peritoneum was closed by a continuous V-Loc suture.

Intraoperative findings, such as endometriosis, adhesions, or cysts, were addressed simultaneously. Histopathological examination was performed at the Institute of Pathology, Bern University Hospital.

### 2.2. Follow-Up

All of the patients had preoperative transvaginal ultrasound imaging at our hospital. Then, three months after the operation, a follow-up was performed either at our hospital or at established gynecologists. During the follow-up, the patients were asked about the resolution of symptoms based on their personal impression (“improvement of symptoms” vs. “no improvement of symptoms”). Postoperatively, we recommended that the patients wait at least three months until achieving pregnancy.

When medical records were incomplete or missing, we performed a telephone survey or sent out a questionnaire to the study participants to collect additional information. The questionnaire asked about symptoms, desire for pregnancy, number of subsequent pregnancies, pregnancy outcome, mode of delivery, and complications.

### 2.3. Outcome Measures

Primary outcome measure:The presence of extrauterine endometriosis and uterine scar endometriosis.Secondary outcome measures:The correlation between uterine scar endometriosis and endometriosis.Surgery outcomes, the improvement of symptoms, the achievement of pregnancy, pregnancy outcome, and complications in patients with and without extrauterine or uterine scar endometriosis.

### 2.4. Data Collection

To investigate the prospects of applying laparoscopic isthmocele repair in the rendezvous technique, we collected baseline characteristics, surgical procedures, and outcomes of all patients from medical records stored in the database of our hospital. The following patient characteristics were collected from the medical records:Patient characteristics: age, body mass index at the date of treatment;Patient’s medical history: number of preoperative pregnancies and deliveries, type of prior cesarean section (elective or emergency), previous surgeries, pre-existing endometriosis;Symptoms that indicate isthmocele surgery: abnormal uterine bleeding, dysmenorrhea, secondary infertility, or uterine scar pregnancy;Surgery characteristics: surgery time, intraoperative blood loss, complications (classified with Clavien–Dindo), suture technique for hysterotomy closure used during the procedure;Outcomes: histologic findings, newly diagnosed endometriosis, postoperative symptoms, pregnancies, and their outcomes.

### 2.5. Statistical Analyses

We used IBM SPSS Statistics (version 25.0, IBM, Armonk, NY, USA) for the statistical analysis. For patient and clinicopathological analyses, we report basic descriptive statistics. In order to compare the characteristics of the different cohorts, the chi-square test, Fisher’s exact test, and ANOVA were used. McNemar’s test was used for paired variables (comparison of pre- and postoperative symptoms). All tests are two-sided, and *p*-values < 0.05 are considered statistically significant.

## 3. Results

### 3.1. Patient Characteristics, Medical History, Symptoms

From April 2004 to March 2020, 83 patients underwent laparoscopic isthmocele resection at the Bern University Hospital. Two patients were lost to follow-up. Descriptive characteristics are presented in Table 1. Seven patients (8%) with pre-existing endometriosis had undergone surgery for endometriosis before the isthmocele correction. Laparoscopy was performed in 10 patients (12%) because of cysts, infertility, ectopic pregnancy, or pain in the past. Five patients (6%) had a recurrent isthmocele. Their prior isthmocele corrections were conducted by laparoscopy, hysteroscopy, or laparotomy.

The most common symptom indicating surgery was secondary infertility (Table 1). In total, 35 patients presented at least two coexisting symptoms: abnormal uterine bleeding and secondary infertility (*n* = 22; 27%), abnormal uterine bleeding and dysmenorrhea (*n* = 10; 12%), and dysmenorrhea and secondary infertility (*n* = 9; 11%).

### 3.2. Surgery Characteristics

For all patients, surgery was performed without major complications. Detailed information is presented in Table 1. The hysterotomy and peritoneum closure was performed in 62 patients (75%) and the hysterotomy closure alone in 20 patients (24%) (missing data: *n* = 1 (1%)). There were no severe complications or excessive bleeding. In two cases, minor complications occurred: a small bladder lesion in one case and the accidental perforation of the uterus in the other. These defects were caught and repaired immediately. After the operation, hematoma of the suprapubic incision was diagnosed in one patient; the patient underwent surgery again, and the bleeding could be stopped. Another patient developed a urosepsis after discharge from the hospital. Readmission to the hospital was followed by antibiotic treatment.

### 3.3. Outcomes

#### 3.3.1. Histological Findings

The most common results were fibrotic or scar tissue. In 11% of patients (*n* = 9), the pathologist detected endometriosis in the scar (also called iatrogenic adenomyosis), defined as the presence of endometrial glands or stromal cells. In six of these patients, extrauterine endometriosis was present at the same time.

##### Endometriosis

Of the 83 patients, endometriosis was detected in 22 (26.5%); 3 of these patients had uterine scar endometriosis only, 13 had extrauterine endometriosis only, and 6 had both types of endometriosis. All of the lesions were confirmed histologically. The total prevalence of endometriosis in the study sample is thus 26.5%. The endometriosis was successfully excised in all but two patients. In one case, there was no follow-up; in the other, there was no desire for pregnancy after the operation. Of the patients with extrauterine endometriosis, 12 patients had peritoneal endometriosis only (63%), 2 had ovarian endometriosis (10.5%), and 3 had deep infiltrating endometriosis only (16%). Two patients had simultaneous ovarian and deep infiltrating endometriosis (10.5%). Of the patients with isthmocele endometriosis, no one had either ovarian or deep infiltrating endometriosis, and just one woman had peritoneal endometriosis in her history (11%). This was the first time that endometriosis was diagnosed in all but one woman. The presence of uterine scar endometriosis was significantly higher in endometriosis patients than in non-endometriosis patients (*p* = 0.004). Details are shown in Table 2. We did not observe more surgery complications when endometriosis was present (χ^2^, *p* = 0.34).

#### 3.3.2. Pregnancy

After surgery, 54 women attempted to get pregnant: 38 women (70.3%) successfully achieved pregnancy, while 15 women (29.6%) failed to get pregnant (one missing value). Ten patients got pregnant two or more times. Of all 55 pregnancies, 33 live births were recorded (60%). The other 22 pregnancies resulted in five ectopic pregnancies (9.1%), nine miscarriages (16.4%), and one abortion (1.8%). In seven cases, the outcome was not reported (12.7%). In the first pregnancies, the live birth rate was 76%; it was 60% in the second pregnancies.

#### 3.3.3. Infertility

The surgery is associated with a significant improvement in fertility (McNemar’s test, *p* < 0.001; see Figure 1). Of all patients with preoperative infertility (*n* = 48), 40 tried to get pregnant; 25 conceived (63%). We could not observe any effect of the suture technique on postoperative fertility (χ^2^, *p* = 0.5882). Comparing extrauterine-endometriosis with non-extrauterine-endometriosis patients, we found no difference in postoperative infertility (χ^2^, *p* = 0.331). Thirty-one of the thirty-seven preoperative infertile patients without extrauterine endometriosis tried to get pregnant, and 19 (61.3%) achieved pregnancy. Nine of the eleven patients with preoperative infertility in the extrauterine-endometriosis group tried to get pregnant, and six (66.7%) achieved pregnancy (see Figure 2). Of those, three had excision of peritoneal endometriosis, two of ovarian endometriosis and deep infiltrating endometriosis (both of the rectovaginal septum and one of the sigma), and one of deep infiltrating endometriosis only (of the rectovaginal septum). Endometriosis was resected completely in all six patients.

Of the women with uterine scar endometriosis and preoperative infertility (*n* = 6, 66.7%), three achieved pregnancy (50%). Those pregnancies resulted in two live births (66.7%) and in the third patient data were missing. Of the patients with other histological findings in the isthmocele and preoperative infertility (*n* = 39, 58.2%), 20 patients became pregnant postoperatively (51%). The pregnancies resulted in 15 live births (75%), 2 miscarriages (10%), 2 ectopic pregnancies (10%), and 1 abortion (5%). Seventeen women remained infertile (44%), and further information was missing in two patients (5%). We could not observe a significant difference in pre- and postoperative infertility in patients with and without uterine scar endometriosis (χ^2^, *p* = 0.9017, and χ^2^, *p* = 1, respectively).

#### 3.3.4. Pregnancy Complications

During pregnancy, seven women (18.4% of all pregnant women; 12.7% of all pregnancies) experienced scar complications: three instances of uterine scar dehiscence, three scar pregnancies (5.5% of all pregnancies), and one placenta previa. We could not find a link between the hysterotomy closure technique (± closure of the peritoneum) and uterine scar complications (excl. peritoneum: *n* = 1; incl. peritoneum: *n* = 6; *p* = 1). The patients with either extrauterine or uterine scar endometriosis had no uterine scar complications.

The cases of uterine scar dehiscence were diagnosed during elective cesarean sections. Preoperatively, one of these patients complained about increasing scar pain.

Cesarean section scar pregnancy was detected in the first trimester and successfully treated with methotrexate in all patients. One patient achieved pregnancy following in vitro fertilization after six miscarriages. The other patient had a previous isthmocele resection in her history and conceived two years after the second isthmocele repair. The third patient got pregnant seven months after surgery. We could not find other causes for cesarean section scar pregnancy.

Compared with the rate of uterine scar pregnancy in the overall population [15], our study shows a rate that is a hundred times higher (5.5% vs. 0.05%).

#### 3.3.5. Abnormal Uterine Bleeding

In the patients included in this study, bleeding disorders improved significantly after surgery (McNemar’s test, *p* < 0.001; Figure 1). After the operation, abnormal uterine bleeding disappeared in 29 patients (64.4%). In extrauterine endometriosis patients, abnormal uterine bleeding decreased as well. We found no difference between the extrauterine-endometriosis and non-extrauterine-endometriosis patients concerning the postoperative bleeding disorders. The suture technique did not show a significant influence on postoperative bleeding disorders (χ^2^, *p* = 0.172).

Preoperative bleeding disorders did not differ between patients with uterine scar endometriosis (*n* = 5, 55.6%) and those with other histological findings (*n* = 36, 47%) (χ^2^, *p* = 1). The difference between postoperative bleeding disorders in the two groups was not statistically significant on the 5% confidence level (χ^2^, *p* = 0.056). The surgery improved bleeding disorders in just 1 patient with (20%) and in 26 patients without uterine scar endometriosis (72%).

Five patients (6%) developed bleeding disorders after the operation: two patients with and three without extrauterine endometriosis, and one patient with and four without uterine scar endometriosis. A reason for these complications might be the recurrence of isthmocele. However, there is no further information available on these cases.

#### 3.3.6. Dysmenorrhea

Patients reported significantly reduced dysmenorrhea after isthmocele repair (McNemar’s test, *p* = 0.03; Figure 1). In women with extrauterine endometriosis, dysmenorrhea improved, but the difference was not significant.

Preoperative and postoperative dysmenorrhea did not differ in patients with and without uterine scar endometriosis (χ^2^, *p* = 0.7583, and *p* = 1)

The suture technique did not affect postoperative dysmenorrhea (χ^2^, *p* = 1.0).

After the operation, five patients (6%) had de novo dysmenorrhea: two patients with and three without endometriosis, no one with uterine scar endometriosis.

## 4. Discussion

The aim of this study is two-fold. First, we set out to investigate whether endometriosis is frequent in patients with isthmocele and whether extrauterine endometriosis and uterine scar endometriosis are correlated. Second, we investigated the outcomes of isthmocele resection in the rendezvous technique.

Regarding the first aim, we find that endometriosis is common among the study population, as it has been detected in every fourth woman. The total prevalence of endometriosis is 26.5%. The literature identified numerous potential causes of endometriosis [16], among them a previous cesarean delivery [17]. Endometrial tissue spreads in the abdomen during surgery. This process could potentially cause endometriosis [7]. Another reason for the higher prevalence in our study population might be the disproportionately high volume of endometriosis patients in our center.

Both endometriosis and isthmocele could cause infertility, and infertility is a possible indication for resection of the isthmocele. In our study, preoperative infertility did not differ in the cohort with or without endometriosis. The infertility rate dropped from 58% preoperatively to 33% in the endometriosis patients and to 39% in the patients without endometriosis after the operation. While the overall pregnancy rate improved considerably due to the surgery, it is unclear whether the improvement of fertility is due to the isthmocele repair or removal of endometriosis in patients with endometriosis. This corresponds to the results of the study by Tsuji et al., which investigates the outcomes of hysteroscopic isthmocele resection. They performed diagnostic laparoscopy and hysteroscopic isthmocele resection in one surgery, diagnosing extrauterine endometriosis in half of the secondary infertile patients. Endometriosis was resected in all cases. The infertility rate after surgery was 26% in patients with endometriosis, similar to our results [12]. Surgical removal of endometriosis improves pregnancy rates [18,19,20]. In our study, it is not clear which part of the combined surgery—resection of endometriosis or isthmocele—has more influence on pregnancy rates in infertile women. Therefore, the simultaneous treatment of endometriosis and isthmocele might be optimal. Surprisingly, the patients with endometriosis (extrauterine as well as uterine scar endometriosis) did not have significantly more dysmenorrhea or other typical endometriosis symptoms. The patients with extrauterine endometriosis showed a longer duration of surgery (143 ± 38 min vs. 120 ± 30 min, *p* = 0.006). Besides the longer duration of surgery, the isthmocele resection could be more challenging for the surgeon in the presence of endometriosis.

In our study, the pathologist detected uterine scar endometriosis (also called iatrogenic adenomyosis) in 11% of the cases. We observed a correlation between extrauterine endometriosis and uterine scar endometriosis. To our knowledge, this is the first time that such an observation has been made. In the literature, uterine scar endometriosis was described several times. Morris detected iatrogenic adenomyosis within the isthmocele in 28% of hysterectomized women with at least one cesarean section in their history; iatrogenic adenomyosis consisted of endometrial glands and stromal cells within the scar [10]. In a few studies, endometriotic lesions in the resected isthmocele were described: Donnez et al. found uterine scar endometriosis in 21% of their patients, Tanimura et al. in 27%, Shapira et al. in 12%, and Fabres et al. in 8%. Simultaneous extrauterine endometriosis was not reported in these cases [8,9,11,21].

Our results highlight the co-occurrence of isthmocele and endometriosis; it is unclear whether or not this is coincidental. However, these results lead to awareness of this subject. The study cannot answer if women with endometriosis are more likely to have an isthmocele.

Regarding the second aim of the study, we demonstrate that laparoscopic isthmocele resection in rendezvous technique leads to a significant improvement of the primary symptom: an increase in the probability of achieving pregnancy and a reduction in abnormal uterine bleeding as well as dysmenorrhea, regardless of the presence of either extrauterine or uterine scar endometriosis. It is encouraging that the pregnancy rate after this procedure in patients with secondary infertility was 63%. These pregnancies resulted in live births in 76% of patients. In the literature, the pregnancy rates after laparoscopic repair are similar to ours. Karampelas et al. published a success rate of 83.3% after laparoscopic isthmocele resection in secondary infertile patients [22]. In other publications, the pregnancy rates ranged from 33% to 80% [8,9,23]. In a recently published meta-analysis, Tanos et al. showed an average pregnancy rate of 72% after hysteroscopic or laparoscopic isthmocele resection [24].

There is evidence that cesarean section reduces fertility by 9% compared with vaginal delivery [25]. The mechanisms of secondary infertility are mostly multifactorial and not completely understood. The accumulation of blood, mucus, or fluid in the isthmocele might impair sperm penetration and, as a consequence, impair fertility [21]. This may explain why the resection of the isthmocele often results in an improvement of fertility.

Besides the observed positive outcomes, laparoscopy led to unexpected consequences in some patients. Specifically, seven patients experienced uterine scar complications during their subsequent pregnancy: there were three cases (5%) of uterine scar pregnancy, three cases (5%) of uterine scar dehiscence, and one case of placenta previa (2%).

In the cases of uterine scar pregnancy, we could not identify a reason, such as a hematoma or complicated surgery. The technique performed was the same as in all the other cases. One patient achieved pregnancy after IVF, the other two spontaneously. In the literature, uterine scar pregnancy after laparoscopic isthmocele resection has not been reported but only described as a pregnancy complication of isthmocele in general [26]. After a cesarean section, uterine scar pregnancy is a potentially dangerous complication, seen with an incidence of 1:2000 [15]. Compared to this relatively low incidence, our study shows a rate that is more than a hundred times higher (1:17). Multiple cesarean sections are considered a risk factor for scar pregnancy, as Zhou et al. reported [27]. In contrast, the three patients in our study had just one cesarean section in their histories. Resection of isthmocele might have a similar effect on wound healing as a repeat cesarean section and therefore increases the risk of scar complications in a subsequent pregnancy. No evidence exists that isthmocele repair might reduce the risk of scar complications during pregnancy [14]. These findings might be relevant for the follow-up of patients after isthmocele surgery.

Uterine scar dehiscence is often detected incidentally during a repeat cesarean section and might be a risk factor for uterine rupture. Bujold et al. described a prevalence of uterine dehiscence of 2.5% after one cesarean section and a prevalence of uterine rupture of 2.4% after one cesarean section following labor [25,28]. Compared with the literature, the prevalence of uterine dehiscence was more than twice as high (5%) in our study.

Laparoscopy is a feasible approach to detect and remove fertility reducing cofactors of isthmocele, such as endometriosis. Besides laparoscopy, hysteroscopy and vaginal repair are the standard surgical methods to treat symptomatic isthmocele. These three approaches all result in similar outcomes and improvement of symptoms in more than 80% of cases. However, while hysteroscopy is a very cost-effective operation with short operative times and hospital stay, laparoscopic or vaginal surgery are associated with longer operative times, more significant intraoperative blood loss, higher costs, and more extended hospital stays and are more challenging for the surgeon [14,29].

However, the choice of surgery is the subject of ongoing controversy. There are recommendations but no guidelines. Vitale et al. recommend the hysteroscopic approach in cases of abnormal uterine bleeding without reproductive desire and a residual myometrial thickness of at least 2.5 mm. Otherwise, the laparoscopic or vaginal repair is indicated to improve the thickness of the residual myometrium, a condition for future pregnancies [14]. Based on the current literature and our findings, we recommend laparoscopic isthmocele repair in patients with secondary infertility even if no other typical endometriosis symptoms are present.

The main limitations of our study are the retrospective setting; the small patient collective; the long study period; the incomplete follow-up and preoperative data, such as sonography records; and the incomplete postoperative data, such as the conception method. In addition, due to the lack of data, we could not consider sonographic myometrium thickness in this study. Unfortunately, due to the retrospective nature of the study, a detailed description or quantification of the pain was not possible.

## 5. Conclusions

Endometriosis is a non-negligible intraoperative finding in patients with symptomatic isthmocele. The laparoscopic approach in the rendezvous technique is safe and effective; therefore, this method should be chosen, especially in women with secondary infertility, and a simultaneous endometriosis resection should be discussed with the patient preoperatively. In follow-up, postoperative pregnancies have to be monitored with care.

## Figures and Tables

**Figure 1 jcm-11-01158-f001:**
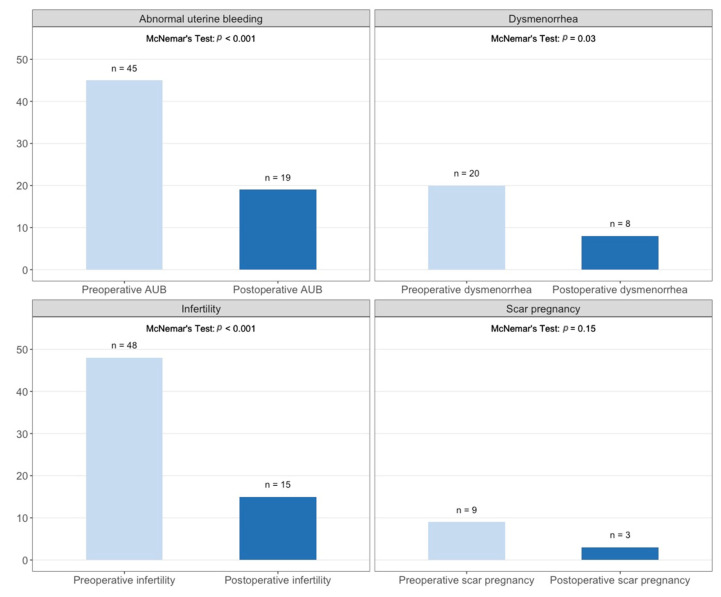
Comparison of symptoms in the study population before and after operation. Significant differences were detected in patients with abnormal uterine bleeding (*p* < 0.001, missing values = 3), dysmenorrhea (*p* = 0.03, missing values = 3), and infertility (*p* < 0.001; missing values = 5). After the operation, five patients developed bleeding disorders and five patients developed dysmenorrhea.

**Figure 2 jcm-11-01158-f002:**
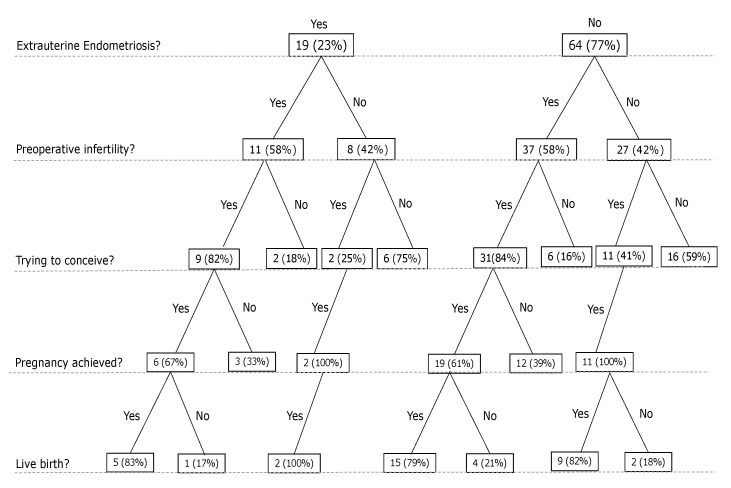
A systematic overview of sample size of patients with and without endometriosis after the operation.

**Table 1 jcm-11-01158-t001:** Descriptive and surgery characteristics of the study population, *n* = 83.

Characteristics	Means ± SD or *n* (%)
Age (years)	34.07 ± 4.3
Body mass index (kg/m^2^)	24.8 ± 4.5
Number of previous cesarean sections	
1	58 (70%)
2	19 (23%)
>2	6 (7%)
Pre-existing endometriosis	7 (8%)
Prior isthmocele correction	5 (6%)
Indication for surgery	
- Abnormal uterine bleeding - Dysmenorrhea - Secondary infertility - Cesarean scar pregnancy	45 (54%) 20 (24%) 48 (58%) 9 (11%)
Surgery time (min)	125 ± 33.3
Blood loss (mL)	42.77 ± 89
Suture technique	
- Hysterotomy closure incl. peritoneum - Hysterotomy closure excl. peritoneum	62 (75%) 20 (24%)
Histological findings of the uterine scar	
- Scar tissue/fibrosis - Endometriosis - Inflammation - Pregnancy tissue	60 (72%) 9 (11%) 5 (6%) 2 (2%)
Extrauterine endometriosis	19 (23%)
Intra-/postoperative complications	4 (5%)

Missing data: suture technique: *n* = 1 (1%), histological findings: *n* = 7 (9%).

**Table 2 jcm-11-01158-t002:** Descriptive characteristics of patients with and without extrauterine endometriosis.

Characteristics	Extrauterine Endometriosis *n* = 19 (23%)	No Endometriosis *n* = 64 (77%)
	Means ± SD or *n* (%)	
Age (years)	34.6 ± 4.2	33.9 ± 4.3
Body mass index (kg/m^2^)	25.2 ± 3.2	24.7 ± 4.88
Previous cesarean section		
1	15 (79%)	43 (67%)
2	3 (16%)	16 (25%)
>2	1 (5%)	5 (8%)
Pre-existing endometriosis	1 (5%)	6 (9%)
Histologic findings in the uterine scar		
Scar tissue/fibrosis	10 (53%)	48 (75%)
Endometriosis	6 (32%) **	3 (5%)
Inflammation	0 (0%)	5 (8%)
Other	0 (0%)	5 (8%)
Unknown	3 (15%)	4 (6%)
Surgery time (min)	143 ± 38 *	120 ± 30
Blood loss (mL)	64 ± 85	42.3 ± 87.8
Indication for surgery		
Abnormal uterine bleeding	10 (53%)	35 (55%)
Dysmenorrhea	6 (32%)	14 (22%)
Secondary infertility	11 (58%)	37 (58%)
Cesarean scar pregnancy	0 (0%)	9 (14%)
Persistence of symptoms/de novo symptoms		
Abnormal uterine bleeding	6 (32%)	13 (20%)
Dysmenorrhea	3 (16%)	5 (8%)
Secondary infertility	3 (16%)	12 (19%)
Cesarean scar pregnancy	0 (0%)	3 (5%)
Revised American Society of Reproductive Medicine (rASRM) score		
rASRM I	14 (74%)	
rASRM II	3 (16%)	
rASRM III	2 (10%)	
rASRM IV	0	

* *p* < 0.01, ** *p* < 0.005; characteristics and outcomes of the endometriosis group (*n* = 19) were compared with those of the non-endometriosis group (*n* = 64). The presence of uterine scar endometriosis is significantly higher (*p* = 0.004) and the surgery time significantly longer (*p* = 0.006) in the endometriosis group.

## Data Availability

The data presented in this study are available on request from the corresponding author.

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
