# Peer review of "Endometriosis and Isthmocele: Common or Rare?"

_jcm, 2022, doi:10.3390/jcm11051158_

Round 1
Reviewer 1 Report
Congratulation for the very interesting paper. Specially the improvements following surgical correction of isthmocele are important for the patients.
Only the idea of a correlation between isthmocele and endometriosis is doubtful. Endometriosis is a very common disease, so a high incidence is quite normal. It is not permissible to compare data from the study collective with the overall population: "The total prevalence of endometriosis is 26.5%, which is more than twice the prevalence of endometriosis in the overall population, which is 10% (16)."
Nevertheless, there are many advantages of the laparoscopic approach to operative correction for isthmocele including resection of endometriosis specially in infertility patients.
Author Response
"Please see the attachment."

Reviewer 2 Report
Thank you for your interesting paper on endometriosis and isthmocele.
I have the following comments:
- regarding your calculation of the the isthmocele/ endometriosis cases, I think you missed to take the 7 missing histological findings into account
if your patient collective includes patients operated in 2004, what was the mean time of follow-up? Did all the patients, even those operated on in 2004, answer all your questionnaires?? How many patients were lost to follow-up? How do you take into account in your calculation the apparently drastic differences in time of follow-up after surgery?
- More information is necessary for the endometriosis patients such as scoring (rAFS and localisation of the lesions - as excision of ovarian endometriosis may also influence fertility) - were all lesions histologically confirmed?
- Table 2: please check formatting of the section "persistence of symptoms"
- were all patients councelled on exision of endometriotic lesions preoperatively?
- Regarding dysmenorrhea, why did you not quantify the pain? (NRS or VAS for example)
- What questionnaires were used in follow-up?
Reviewer 3 Report
The authors analyzed a possible correlation between isthmocele and
endometriosis and to investigate the outcome of laparoscopic isthmocele resection in the rendezvous technique. In fact, endometriosis is often found in the isthmocele, however, the exact correlation has not been clarified yet. So, this study might be valuable, however, there are major faults below.
The title of this study is "Endometriosis and isthmocele: common or rare?", however, half of this manuscript describe things that are not related to endometriosis, i.e., the outcome of laparoscopic isthmocele resection in the rendezvous technique. This makes difficult for readers to understand the meaningful of this study. The authors set up three primary outcome measures, but they are too many. I suggest primary outcome of this study should be focused on the matter regarding endometriosis. The authors should investigate which type of endometriosis, i.e. superficial peritoneal lesion, endometrioma, or deeply infiltrating endometriosis is correlated to isthmocele. The authors should also discuss whether extrauterine endometriosis is cause or consequence of isthmocele.
The authors should describe diagnostic criteria for isthmocele in this study.
Round 2
Reviewer 3 Report
Thank you for revising your manuscript. The content has been improved, but further improvements are required.
Regarding point 1, you changed the description of primary and secondary outcome measures. Primary outcome is reasonable. I suggest secondary outcome should also be focused on endometriosis, for example like below.
・The correlation between isthmocele and endometriosis
・Surgery outcomes, the improvement of symptoms, the achievement of pregnancy, pregnancy outcome and complication in patients with and without uterine scar endometriosis.
It is really intriguing how different in postoperative pregnancy outcomes and complication between uterine scar endometriosis and other isthmocele. Authors compared extrauterine endometriosis and no endometriosis, it is good to analyze the correlation between isthmocele and endometriosis. However, when analyzing postoperative outcomes like symptom improvement and pregnancy outcomes, it is better to compare uterine scar endometriosis and other isthmocele (no endometriosis in uterine scar).
Regarding point 2,3,4, manuscript is well revised.
